# Production of 3D Printed Bi-Layer and Tri-Layer Sandwich Scaffolds with Polycaprolactone and Poly (vinyl alcohol)-Metformin towards Diabetic Wound Healing

**DOI:** 10.3390/polym14235306

**Published:** 2022-12-05

**Authors:** Sena Harmanci, Abir Dutta, Sumeyye Cesur, Ali Sahin, Oguzhan Gunduz, Deepak M. Kalaskar, Cem Bulent Ustundag

**Affiliations:** 1Center for Nanotechnology & Biomaterials Application and Research (NBUAM), Marmara University, Istanbul 34722, Turkey; 2Department of Bioengineering, Faculty of Chemical and Metallurgical Engineering, Yildiz Technical University, Istanbul 34210, Turkey; 3UCL Division of Surgery and Interventional Sciences, Royal Free Hospital Campus, London NW3 2PF, UK; 4Department of Biochemistry, Faculty of Medicine, Marmara University, Istanbul 34722, Turkey; 5Department of Metallurgical and Materials Engineering, Faculty of Technology, Marmara University, Istanbul 34722, Turkey; 6Health Biotechnology Joint Research and Application Center of Excellence, Esenler, Istanbul 34220, Turkey

**Keywords:** metformin, diabetic wound healing, drug release, 3D-printed scaffold, sandwich scaffolds

## Abstract

Type 2 diabetes mellitus (T2DM) is a chronic disease characterized by impaired insulin secretion, sensitivity, and hyperglycemia. Diabetic wounds are one of the significant complications of T2DM owing to its difficulty in normal healing, resulting in chronic wounds. In the present work, PCL/PVA, PCL/PVA/PCL, and metformin-loaded, PCL/PVA-Met and PCL/PVA-Met/PCL hybrid scaffolds with different designs were fabricated using 3D printing. The porosity and morphological analysis of 3D-printed scaffolds were performed using scanning electron microscopy (SEM). The scaffolds’ average pore sizes were between 63.6 ± 4.0 and 112.9 ± 3.0 μm. Molecular and chemical interactions between polymers and the drug were investigated with Fourier transform infrared spectroscopy (FT-IR) and X-ray diffraction (XRD). Mechanical, thermal, and degradation analysis of the scaffolds were undertaken to investigate the physico-chemical characteristics of the scaffolds. Owing to the structure, PCL/PVA/PCL sandwich scaffolds had lower degradation rates than the bi-layer scaffolds. The drug release of the metformin-loaded scaffolds was evaluated with UV spectrometry, and the biocompatibility of the scaffolds on fibroblast cells was determined by cell culture analysis. The drug release in the PCL/PVA-Met scaffold was sustained till six days, whereas in the PCL/PVA-Met/PCL, it continued for 31 days. In the study of drug release kinetics, PCL/PVA-Met and PCL/PVA-Met/PCL scaffolds showed the highest correlation coefficients (R^2^) values for the first-order release model at 0.8735 and 0.889, respectively. Since the layered structures in the literature are mainly obtained with the electrospun fiber structures, these biocompatible sandwich scaffolds, produced for the first time with 3D-printing technology, may offer an alternative to existing drug delivery systems and may be a promising candidate for enhancing diabetic wound healing.

## 1. Introduction

Wounds occur due to the deterioration of the epithelial integrity of the skin through different reasons such as burns, diseases, and trauma [1]. Chronic wounds are unable to undergo the complete natural healing process as they remain in an inflammatory stage owing to disruption in the healing process [2,3]. Chronic wounds exhibit higher levels of bacterial presence, inflammatory cytokines, reactive oxygen derivatives, and protease activity in the wound microenvironment as compared to normal wounds [4]. Chronic wounds, classified into three main groups, namely, pressure ulcers, venous ulcers, and diabetic foot ulcers, are observed more commonly in diabetic and obese population groups [5].

Diabetes mellitus (DM) is a chronic disease, characterized by hyperglycemia and dysfunction of insulin secretion and sensitivity, which has shown a steep increase in incidence and prevalence each year. Poorly managed DM results into various micro and macro complications. Diabetic wounds are known to be one of the vivid macro complications of DM [6]. Long-term hyperglycemia adversely affects the normal wound healing process, causing wounds to become chronic [7]. These wounds are more difficult to heal, and a more complex treatment is required to achieve complete healing. Amongst many surgical and non-surgical methods for healing of chronic wounds, one of the most common methods is wound dressings [8].

Wound dressings protect the wound area to allow formation of neo-tissue and blood vessels. Common wound dressing types are skin grafts, films, scaffolds, nanofiber meshes, and hydrogels [9]. As compared to different types of wound dressings, scaffolds are advantageous in terms of their capability to mimic natural ECM components, highly interconnected porous structure allowing room for enhancing cell adhesion, integration and further differentiating into targeted tissue phenotypes allowing controlled degradation and containing bioactive agents that will contribute to wound healing [10]. An ideal scaffold should be biocompatible, biodegradable, non-toxic, non-immunogenic, with suitable physical and mechanical properties, and have a degradation rate that follows the wound healing process [11].

Many different methods can produce scaffolds with the desired properties in wound healing, but 3D-printing technology is a fabrication technique that has been increasingly used in recent years in development of patient-specific tailorable hard and soft tissue analogues [12,13]. Scaffolds designed with computer-aided drafting software can be fabricated in a layerwise bottom-up approach using 3D-printing technology with several types of biomaterials and live-cell-based bioinks [14,15,16,17]. Synthetic polymers, such as polyglycolic acid (PGA), polylactic acid (PLA), polyethylene glycol (PEG), polyvinyl acid (PVA), polycaprolactone (PCL), and natural polymers, such as, fibrin, fibronectin, gelatin, collagen, alginate, and chitosan, are commonly used biomaterials for wound dressing applications [18].

Poly (vinyl alcohol) (PVA) is a frequently used synthetic polymer in tissue engineering, wound dressing, drug delivery systems, as well as in preparing artificial corneas and contact lenses because of its biocompatible, biodegradable, non-toxic properties [19]. PVA provides sufficient mechanical and chemical stability, flexibility, and permeability in biomedical applications [20,21]. This hydrophilic polymer’s capability to form solid hydrogen bonds with water molecules could enhance wound healing as it makes room for the moisture around the wound area [22,23]. Moreover, the high swelling characteristic of PVA helps it mimic natural tissues and be accepted into the body [24]. PVA is a polymer that can degrade in a short time [25]. Therefore, PVA can be blended with other polymers, and organic and inorganic materials to reinforce chemical and mechanical properties [26]. Polycaprolactone (PCL) is a biocompatible, non-toxic synthetic polymer [27]. Its efficient drug permeability makes it a good candidate for tissue engineering and drug release applications [28]. As compared to other polymers, the slow biodegradable nature of PCL yields a higher mechanical stability with desired controlled release and long-term degradation profiles [29].

Metformin (Met) is a biguanide derivative drug used to treat type 2 diabetes. It has an antihyperglycemic effect that increases glucose production and insulin sensitivity [30]. Beyond that, Met has antioxidant, anti-fibrotic, and anti-proliferative effects, and it can increase the quantity and function of endothelial precursor cells. Metformin may promote wound healing due to its immunomodulatory, regulation of various inflammation pathways, and anti-inflammatory properties [31]. Han et al. applied metformin treatment to diabetic rats. Results showed that Met increased wound healing and angiogenesis in diabetic rats [32]. Zhao et al. applied Met, resveratrol, and rapamycin locally for 14 days to rats that had cutaneous wounds and observed that metformin showed promising results in wound healing, epithelization, number of hair follicles, and collagen deposition [33].

In the present study, 3D-printed metformin-loaded scaffolds with different layerwise constructions were designed, fabricated, and characterized, aiming at healing of diabetic wounds. To the best of the authors’ knowledge, the majority of earlier literature was based on different material composition [13,34] as well as fabrication methodologies such as electrospinning, etc. [34,35,36,37], aiming at hard tissue engineering applications [36,37]. The novelty of the present study lies in development of a 3D-printed composite hybrid structure, capable of a sustained long-time drug release profile targeted towards diabetic wound healing. PCL/PVA, PCL/PVA-Met and PCL/PVA/PCL, and PCL/PVA-Met/PCL sandwich scaffolds were successfully fabricated and preliminarily tested with human fibroblast cells for their cytocompatibility.

## 2. Materials and Methods

Metformin was used for its anti-inflammatory effects, which is an important property in wound healing. The morphological, physicochemical, and mechanical characteristics of the scaffolds were investigated, followed by drug release kinetics and in vitro cytotoxicity assessments. The methodology of the development and characterization of the scaffolds is presented briefly in the following.

### 2.1. Materials

Metformin (Met), polycaprolactone (PCL, Mw: 80 kDa), poly (vinyl alcohol) (PVA, Mw 89–98 kDa, 99% hydrolyzed), and chloroform were procured from Sigma-Aldrich, USA. The HFF-1 human fibroblast cells were obtained from American Type Culture Collection (ATCC), and standard sterile cell culture techniques were undertaken for all in vitro assessments. DMEM, penicillin/streptomycin, and 10% fetal bovine serum (FBS) were procured from Gibco (Thermo Fisher, Waltham, MA, USA). The DAPI (4′,6-diamidino-2-phenylindole) staining kit was obtained from Thermo-Invitrogen.

### 2.2. Methods

#### 2.2.1. Composition of the Polymer Solutions

PCL, PVA, and PVA-Met solutions were prepared separately according to the polymer contents, as described in Table 1. PCL pellets were dissolved at 20% *w/v* in chloroform at room temperature (RT) for 2 h. PVA powder was dissolved in distilled water at 20% *w/v* at 100 °C for 1.5 h. Metformin was added to the PVA solution at 3% *w/v* concentration, and the solution was stirred at RT for 30 min.

#### 2.2.2. Design and Fabrication of the 3D-Printed Scaffolds

The scaffolds were designed in Solidworks (Dassault Systèmes SolidWorks Corporation, Waltham, MA, USA), and further processed for 3D printing in Simplify 3D slicing software. In the present work, four types of scaffold architecture, (a) design 1: 6 layers PCL/3 layers PVA, (b) design 2: 3 layers PCL/3 layers PVA/3 layers PCL, (c) design 3: 6 layers PCL/3 layers metformin-loaded-PVA, and (d) design 4: 3 layers PCL/3 layers metformin-loaded-PVA/3 layers PCL, were designed (Figure 1). The designed scaffolds were fabricated using an extrusion-based 3D printer (Hyrel 3D, SDS-5 Extruder, Norcross, GA, USA). Prior to printing, two 10 mL syringes were individually filled with PCL, PVA, and Metformin-loaded PVA solutions. The scaffolds were printed as square blocks, having a dimension of 20 mm × 20 mm × 1 mm. The following optimized parameters were finalized for printing all the designed scaffolds throughout this study, infill density: 96%, no. of layers: 9, infill pattern: linear, flow rate during the printing process: 1 mL/h, and printing speed at 10 mm/s.

#### 2.2.3. Rheological Characterization

The physical parameters of PCL, PVA, and PVA-Met solutions, including density and viscosity, were investigated at RT. A 10 mL standard density bottle and DIN ISO 3507-Gay-Lussac (Boru Cam Inc., Istanbul, Turkey) were used to analyze the density of the solutions. The viscosity of the solutions was evaluated using a digital viscometer (RM 200 Plus, Lamy Rheology, Champagne au Mont d’Or, France).

#### 2.2.4. Scanning Electron Microscopy (SEM)

The surface morphology of the fabricated scaffolds was examined under a scanning electron microscope (EVA MA 10 ZEISS, Jena, Germany). The samples were gold (Au) and palladium (Pd) coated in a sputter coating machine (Quorum SC7620, Quorum Technologies Ltd., East Sussex, UK) for 120 s. The average pore sizes of each scaffold were measured using Analysis5, Olympus (Münster, Germany) software.

#### 2.2.5. Fourier Transform Infrared (FTIR) Spectroscopy

Molecular and chemical interactions of the scaffolds were investigated using Fourier transform infrared spectroscopy (FTIR, FT/IR 4700, Jasco, Tokyo, Japan). All the spectra were taken at wavelengths between 450–4000 cm^−1^ and at 4 cm^−1^ resolution.

#### 2.2.6. X-ray Diffraction Analysis (XRD)

The crystalline phases of PCL, PVA, and metformin present in the 3D-printed scaffolds were examined by performing X-ray diffraction (XRD) analysis (Shimadzu XRD-6100, Kyoto, Japan; Cu-Kα radiation at λ = 1.54060 A°; 40 mV and current at 30 mA; 2θ range of 10–90°; scan speed 2°/min).

#### 2.2.7. Thermal Analysis

Differential scanning calorimetry (DSC, Shimadzu, Kyoto, Japan) was used to evaluate the thermal behavior of the scaffolds. The analysis was performed in a closed pan, at a temperature range of 35–300 °C, and a heating rate of 10 °C/min.

#### 2.2.8. Degradation Characteristics

Before the test, the initial weights (W_0_) of the scaffolds were measured and recorded, as in the wet weight analysis. In the degradation test, the scaffolds in Eppendorf tubes were removed from the phosphate buffered saline (PBS, pH 7.4) and dried at room temperature for 24 h, and each scaffold was weighted (W_t_) every two days for 25 days. PBS was used to mimic the inorganic phase of human plasma. The degradation indices (D_i_) for each scaffold were calculated according to Equation (2) [38]:(1)$$D_{i}=\left(W_{0}-W_{t}\right)/W_{0}\times 100,$$

#### 2.2.9. Drug Loading Study

In the in vitro drug release kinetics analysis, firstly, the metformin solution was prepared at 6 different concentrations (4, 6, 8, 10, 12, and 14 μg/mL) with PBS to create a calibration curve (Appendix A, please see in Appendix A). The PCL/PVA-Met and PCL/PVA-Met/PCL scaffolds were cut into 10-mg pieces and placed into the Eppendorf tubes. Then, 3 mL of PBS solution (pH 7.4 at 37 °C) was added to the tubes and then they were kept in a shaking incubator (37 °C, 200 rpm) till the end of the test. Furthermore, 3 mL of PBS was taken from the tubes at certain hours and measurement was made with a UV-vis spectrophotometer (Shimadzu UV-3600, Kyoto, Japan) at 233 nm. Afterward, 3 mL of fresh PBS was added to the tubes to maintain the release. The test lasted for a month.

Encapsulation efficiency (EE) is used to measure the rate of the drug successfully loaded onto the scaffolds, and this rate is determined by a standard assay procedure. Metformin-loaded scaffolds were weighed at 10 mg and stirred continuously to ensure complete dissolution in their organic solvents in the beaker. Next, 3 mL from each solution was taken and the metformin content was determined with a UV-visible spectrophotometer (Shimadzu UV-3600, Kyoto, Japan) at 233 nm. Measurements were made three times. The EE% was calculated according to Equation (3) [39]:(2)$$\mathrm{EE}\%=\frac{\mathrm{The}\;\mathrm{mass}\;\mathrm{of}\;\mathrm{actual}\;\mathrm{drug}\;\mathrm{loaded}\;\mathrm{in}\;\mathrm{scaffolds}}{\mathrm{Mass}\;\mathrm{of}\;\mathrm{drug}\;\mathrm{used}\;\mathrm{in}\;\mathrm{scaffolds}\;\mathrm{fabrication}},$$

#### 2.2.10. In Vitro Drug Delivery Study

The drug release kinetics of the scaffolds were evaluated with Korsmeyer–Peppas, zero order, first order, Higuchi, and Hixson–Crowell mathematical models. Mathematical model equations of Korsmeyer–Peppas (3), zero-order (4), first order (5), Higuchi (6), and Hixson–Crowell (7) are as below, respectively [6]:Q = *Kt^n^*,(3)
Q = *K_0_t*,(4)
In (1−Q) = −*K_1_t*, (5)
Q = *K_h_t*^1/2^, (6)
Q^1/3^ = *K_hc_t*(7)
where Q is the fractional amount of drug release at time *t*; *K*, *K_0_*, *K_1_*, *K_h_*, and *K_hc_* are the kinetic constants for the models; and n represents the diffusion constant, which is an indicator of the drug release mechanism.

#### 2.2.11. In Vitro Cytocompatibility Assessment

The human fibroblast cell line (HFF-1) was used to investigate the response of cells to the scaffolds with different compositions. Firstly, scaffolds in the 96-well plates were sterilized using ultraviolet (UV). They were incubated for an hour in the DMEM that contained FBS (10% *v/v*), 0.1 mg/mL penicillin/streptomycin in a 37 °C and 5% CO_2_ incubator (Sanyo, Akribis Sci. Ltd., Knutsford, UK). Then, scaffolds were taken out, and the excess medium was separated. Into the plates, 5 × 10^3^ cells/mL fibroblast cells were then seeded. The scaffolds were incubated in a 37 °C and 5% CO_2_ humidified incubator for a week. Meanwhile, monolayer (2D) cell cultures were incubated with the same number of cells at 200 µL under the same conditions for cross-control. The cytocompatibility of the scaffolds was determined with a cytotoxicity detection kit (MTT from Glentham Life Sciences, Corsham, UK) at the end of the first, third, and seventh days. The ELISA reader (Perkin Elmer, Enspire, Waltham, MA, USA) at 570 nm wavelength was used to examine the absorbance values of the MTT test. The test was performed in triplicate, and the average of the results was considered for the analysis.

DAPI staining was performed to examine the fibroblast cell attachment to the scaffolds. At the end of the first, third, and seventh day, scaffolds were removed from the growth medium and washed with PBS. Then, they were kept with 4% formaldehyde for 30 min at room temperature and rewashed with PBS. For staining, 1 µg/mL DAPI (Invitrogen, Waltham, MA, USA) was added to each scaffold and incubated for 20 min. The scaffolds were taken from the DAPI solution and placed between slides and investigated by using an inverted fluorescence microscope (Leica DM IL LED Microsystems, Leica Microsystems, Wetzlar, Germany).

The cell morphology of the fibroblasts was investigated by using a scanning electron microscope (SEM). The scaffolds were removed from the growth medium after 1, 3, and 7 days and exposed to 4% glutaraldehyde. They were dehydrated through serial dilutions of ethanol and dried in air. Later, the scaffolds were sputter-coated with gold-palladium and analyzed under an SEM (EVO MA-10, Zeiss, Jena, Germany).

#### 2.2.12. Statistical Analysis

A one-way ANOVA analysis program was used to carry out the statistical analysis. The pore size measurement was performed with SPSS 17.0 analysis software. The significance level was taken as *p* < 0.05, and the data were labelled for *p* < 0.05 with (*), *p* < 0.01 (**), and *p* < 0.001 (***). All data were presented with their means ± standard deviations.

## 3. Results and Discussion

Polymer solutions must meet certain material and biological criteria along with optimal room to tailor the printability to successfully fabricate the scaffolds. Printability is important in two ways; the first, it directly affects the processability of materials, and ensures structure stability during the printing process [40]. Viscosity and density also have a direct impact on the printability of the solution. Viscosity affects the droplet size coming from the needle tip and the flow of the solution. The polymer solution should be viscous enough to print the material layer-by-layer but not so viscous as to clog the needle tip [41]. The viscosities of the PCL, PVA, and PVA-Met solutions was thus analyzed separately. At the shear rate 50 s^−1^, the viscosities of PCL, PVA, and PVA-Met solutions were 4129.4, 1870.5, and 1966.6 mPa.s, respectively. The addition of metformin increased the viscosity of the solutions. In Cesur et. al.’s study, the viscosity of the Met-loaded PVA solution was also higher than the PVA solution [30]. The densities of the PCL, PVA, and PVA-Met were also investigated. The PCL solution has the highest density value with 1.4064 g/m^3^. By adding metformin to the PVA solution, the density of the solution increased slightly from 1.0325 to 1.0384 g/m^3^.

Surface morphology of the 3D-printed scaffolds and their pore architecture are one of the crucial parameters in determining their influence in wound healing aspects in tissue engineering. Representative SEM images and the average pore sizes of the novel 3D-printed scaffolds are shown in Figure 2. The average pore sizes of PCL/PVA (Figure 2A,E) and PCL/PVA/PCL (Figure 2B,F) scaffolds were found to be 112.9 ± 3.0 and 84.1 ± 2.7 μm, respectively. The addition of metformin caused a decrease in the pore sizes of 3D scaffolds to 82.9 ± 4.0 for PCL/PVA-Met (Figure 2C,G) and 63.6 ± 4.0 μm for PCL/PVA-Met/PCL (Figure 2D,H). The report by Morawska et al. established that the addition of drugs causes a reduction in the pore sizes of the scaffolds [42]. Thus, PCL/PVA/PCL and PCL/PVA-Met/PCL sandwich scaffolds have lower pore sizes when compared with the bi-layered structure owing to their design. The surface morphology of the scaffolds was similar, but the sandwich scaffolds’ homogeneity was higher as compared to the others; this could be the result of the highly crystalline structure of PCL in the upper layer. All scaffolds’ pore size diameters were within a range of 63.6 ± 4.0–112.9 ± 3.0 μm, providing the necessary fibroblast attachment and skin regeneration to increase wound healing.

Optimal open porosity, controlled pore size, and interconnected pore structure in the scaffolds are necessary to support nutrient and gas transfer, cell adhesion, differentiation, and proliferation, as well as vascularization towards neo-tissue formation [43,44,45]. Since the porosity directly affects the mechanical properties, the porosity of the scaffolds should be compatible with the host tissue and possess mechanical stability to provide the necessary tissue growth in the microenvironment [46,47]. Ideal pore sizes differ according to the type of tissue and cell. The pore size of 50–160 μm is optimal for the attachment of fibroblasts [48]. Henkel et al. revealed that for skin regeneration, 3D scaffolds should possess an average pore size between 20 and 125 μm [49]. In tissue engineering applications, enhanced cell growth and ECM production could be achieved at the porosity of the scaffolds reaching ~90% [50].

Molecular and chemical interactions of the scaffolds and the success of the loading of Met in the scaffolds were investigated by conducting FTIR analysis. Figure 3A shows the FTIR spectra of Met, PCL, PVA, and 3D-printed scaffolds. In Figure 3A(a), Met has two characteristic peaks of N-H primary stretching at 3366 and 3289 cm^−1^ [51]. N-H stretching and the N-H bending vibration peaks of the primary amine group were observed at 3147 and 1539 cm^−1^, respectively. At 1621 cm^−1^ there were C=O stretching/C-N stretching/N-H bending bands. The asymmetric bending vibration band of the CH_3_ group was at 1471 cm ^−1^ [6]. In Figure 3A(b), for PVA, a large band of O-H stretching vibration of the hydroxyl group due to the strong inter and intramolecular interactions of hydrogen bonds are observed at 3171 cm^−1^. The bands at 2909 and 850 cm^−1^ are assigned because of the C-H symmetric stretching and C-H rocking bands of alkyl groups. The peaks of C=O and C-O-C stretching of acetate groups are seen at 1712 and 1239 cm^−1^, respectively [38,52]. FTIR spectra of PCL are shown in Figure 3A(c). Asymmetric and symmetric CH_2_ stretching bands were at 2940 and 2865 cm^−1^, respectively. The sharpest band was at 1722 cm^−1^ due to C=O stretching of the carbonyl group. C-O-C asymmetric and symmetric stretching were observed at 1238 and 1165 cm^−1^ [53]. Unlike the morphological structures of the scaffolds, there is not much difference in their chemical structures. Because of the higher amount of PCL in the scaffolds, the nature of the survey lines bear closer resemblance to that of the PCL, rather than the ones corresponding to the PVA. Some of the characteristic peaks of metformin were observed in drug-loaded scaffolds. N-H stretching and bending peaks due to the primary amine group are seen at 3366, 3289, 1621 and 1539 cm^−1^. However, since metformin was loaded into the PVA solution, and the amount of drug contained in the scaffolds was very low, sharp peaks were not observed, and shifts were also visible. These shifts might be present due to intermolecular interactions between PVA and metformin. As a result, it was revealed that metformin was successfully loaded into the scaffolds.

The crystalline structure of the scaffolds was determined by performing XRD analysis. Diffraction patterns of PCL, PVA, metformin, PCL/PVA, PCL/PVA/PCL, PCL/PVA-Met, and PCL/PVA-Met/PCL scaffolds are shown in Figure 3B(a–g). Due to the semi-crystalline structure of PCL, the sharp diffraction peaks at 2θ = 21.56°, 38.42°, 44.65°, 64.95°, and 78.02° were observed (Figure 3B(a)) [54,55]. PVA has a semi-crystalline structure due to hydrogen bonding between hydroxyl groups [56]. For PVA, sharp peaks were observed at 2θ = 38.1° and 44.32° while broad peaks were observed at 2θ = 19.64°, 20.78°, and 22.78° (Figure 3B(b)) [56,57,58]. Because of its crystalline nature, metformin showed sharp peaks at 17.67°, 22.38°, 23.26°, 24.54°, 29.5°, 31.27°, 37.14°, and 45.86° (Figure 3B(c)) [51,59]. The characteristic peaks of PCL and PVA were observed in all scaffolds. The structural difference in PCL/PVA and PCL/PVA/PCL scaffolds did not cause a significant difference in the crystalline characteristics of the scaffolds. It was observed in the diffractograms that both scaffolds gave peaks at similar angles. In Met-loaded scaffolds, sharp peaks of Met were observed in a lower intensity and broader manner due to the relatively low drug concentration in the solution. No significant shift was observed in the angles of the diffraction peaks of the pure and drug-loaded scaffolds. XRD results supported the observations during FTIR, thus providing confidence in the fact that the drug was effectively loaded into the scaffold.

The thermal properties of neat and drug-loaded scaffolds with different designs were investigated by conducting DSC analysis. Figure 3C represents the DSC thermograms of PCL/PVA, PCL/PVA/PCL, PCL/PVA-Met, and PCL/PVA-Met/PCL scaffolds. PCL exhibited a sharp melting peak at 60 °C, while the melting point of pure PVA was observed at ∼223 °C [60]. DSC results showed that each scaffold had the melting peak of PCL crystals at ∼58 °C; however, these peaks are not as sharp as in pure PCL due to the inherent composite structure. Similarly, in the DSC analysis of PCL/PVA fibers, reported by Agarwal et al., a wider and shallow peak was observed at 60 °C as compared to those of pure PCL peak [61]. The absence of the melting peak of PVA in the thermograms could be the result of considerably lower ratio of PVA in the scaffolds as compared to PCL, and the significant decrease in crystal fraction due to the interaction of PVA with metformin [62]. The sharp endothermic peak of metformin at 232 °C was observed as a smaller and wider peak due to the small amount of drug in the PCL/PVA-Met and PCL/PVA-Met/PCL scaffolds. On the contrary, metformin-loaded PLA microparticles, reported in Bouriche et al., did not show a sharp peak of metformin due to the interaction of hydrophilic metformin with PLA [59]. Overall, there were no significant differences between neat and drug-loaded scaffolds, and all the scaffolds showed the appropriate thermal stability as the melting temperatures showed a correlation with pure PCL, PVA, and metformin.

Degradation is a process in which the structures are exposed to PBS dissolve over time and the mass of the structure decreases [63]. Since the degradation profiles of the materials will affect the mechanical properties of the scaffolds, it is crucial for the scaffolds to have the appropriate degree of degradation to provide necessary mechanical support during the entire period of formation of neo-tissue [64]. The degradation profile of the scaffolds was examined for 25 days in PBS (pH 7.4) to mimic body fluid (Figure 4). As a result of the 25-day analysis, the PCL/PVA/PCL scaffold had the lowest degradation rate, while the PCL/PVA-Met scaffold had the highest degradation rate. Biomaterials with higher water absorption capacity are known to degrade faster, thus PCL/PVA and PCL/PVA-Met scaffolds might have degraded faster as their swelling capacity was higher than PCL/PVA/PCL and PCL/PVA-Met/PCL scaffolds [65]. The tri-layered PCL/PVA:SA/PCL fibers also had a higher degradation rate than the PCL/PVA:SA [66]. In this study, the addition of metformin resulted in a decrease in the tensile strength and an increase in the swelling ability of the scaffolds.

The release of metformin from PCL/PVA-Met and PCL/PVA-Met/PCL scaffolds was investigated in a PBS solution at pH = 7.4 and 37 °C. The UV spectra of metformin were obtained by measuring the absorbance of six different metformin concentrations from 4 to 12 μg/mL in a UV-Vis spectrophotometer (Figure 5A). A standard calibration curve of metformin (Appendix A) was constructed using the absorbance values measured at 233 nm against different concentrations (R^2^ = 0.9898). The encapsulation efficiency (EE) of the metformin loaded into the scaffolds was also investigated. Figure 5B showed that the %EE of the PCL/PVA-Met and PCL/PVA-Met/PCL scaffolds were 46.3 ± 2.3 and 48.8 ± 3.4%, respectively. The drug release profiles of the scaffolds are presented in Figure 5C,D. According to the results, in the early hours of the release study, PCL/PVA-Met scaffolds showed a burst release of 50.3%, while PCL/PVA-Met/PCL scaffolds only released 32.4% of the drug. It was observed that PCL/PVA-Met scaffolds released 71.9% of the drug and PCL/PVA-Met/PCL scaffolds 51%, at the post-completion evaluation of the first 24 h release profile. Each scaffold released the drug gradually over time and drug release in PCL/PVA-Met and PCL/PVA-Met/PCL scaffolds had reached up to 100% at 144 and 744 h, respectively. In a study by Chogan et al., it was found that three-layer PCL:Cs/PVA+Met/PCL:Cs fiber prevented the burst release of the drug as compared to the single PVA mats; furthermore, the three-layer scaffolds showed sustained release of the drug over 15 days [67]. In this study, Met was loaded onto PVA due to its water solubility. Both scaffolds showed sustained release of the drug despite the high hydrophilicity and non-crosslinking of PVA, but sandwich scaffolds released the drug more slowly as was expected. This might be attributed to the innate nature of the PCL, which has lower degradability in the upper and lower layers, providing controlled release of the drug. Met-loaded sandwich scaffolds can release the drug in approximately one month, providing drug delivery to that area during wound healing and thus promoting healing.

The Korsmeyer–Peppas, zero order, first order, Higuchi, and Hixson–Crowell models were used to investigate the release kinetics of metformin (Figure 6). The correlation coefficients (R^2^) and the kinetic constants were obtained by applying the drug release data of metformin-loaded scaffolds to the indicated models (Table 2). The correlation coefficient (R^2^) determines the most appropriate release pattern. In this study, PCL/PVA-Met and PCL/PVA-Met/PCL showed the highest R^2^ values of 0.8735 and 0.889 for the first order release model, respectively.

In the Korsmeyer–Peppas model, the value of n represents the drug release mechanism. The ranges of n values are given in Table 3 [68]. The n value was calculated for each scaffold, and diffusion mechanisms of the drug release of the corresponding scaffolds were determined. The results showed that the release of metformin from both scaffolds followed the super case II transport mechanism.

There is a distinct difference between PCL/PVA-Met and PCL/PVA-Met/PCL scaffolds in drug-release studies. The drug release from PCL/PVA-Met and PCL/PVA-Met/PCL (sandwich scaffold) was 71.9% and 51% in 24 h, respectively. The drug release in the PCL/PVA-Met scaffold ended at six days, whereas in the PCL/PVA-Met/PCL it continued for 31 days. This is because PVA, which dissolves well in water, was printed between the PCL layers, which takes a long time to degrade.

Biocompatibility of a material refers the biomaterial or scaffolds’ ability to not cause an immune response and toxic effect within the host tissue sites [69]. In this study, cytotoxicity and cell viability of all scaffolds were evaluated by MTT analysis. The MTT assay was performed at 1, 3, and 7 days of incubation to examine the effect of pure and drug-loaded scaffolds with different designs for cell proliferation in human fibroblast cells. The results of the cell viability test showed that all scaffolds had a lower viability and proliferation rate than the control group (Figure 7). According to the results, at the end of the first day, the highest cell viability was observed in PCL/PVA/PCL (83.1%), and the lowest was observed in PCL/PVA-Met (65.9%). On day three, cell viability and proliferation increased in PCL/PVA (71.6%) and PCL/PVA-Met (73.6%) scaffolds, while a decrease occurred in PCL/PVA/PCL (69.5%) and PCL/PVA-Met/PCL (69.1%) scaffolds. At the end of seven days of fibroblast incubation, the viability values of PCL/PVA, PCL/PVA/PCL, PCL/PVA-Met, and PCL/PVA-Met/PCL scaffolds were 89.3, 89.2, 91.8, and 92.8%, respectively. Excessive fibroblast proliferation in wound healing can cause fibrosis and keloids, and thus the produced scaffolds should not trigger excessive fibroblast proliferation and cause scar tissue formation in order to improve wound healing [70]. As a result of MTT analysis, the structural difference between the scaffolds did not make a difference to cell growth. In contrast, the addition of metformin to the scaffolds increased fibroblast viability and proliferation, but this increment was not very significant.

The material-cell interaction and the morphology of the fibroblasts on the scaffolds were investigated by SEM analysis on the seventh day of incubation. Figure 8 shows that fibroblast attachment and proliferation on all of the scaffolds were sufficient. However, cell adhesion and proliferation were more enhanced in PCL/PVA-Met and PCL/PVA-Met/PCL scaffolds than in PCL/PVA and PCL/PVA/PCL scaffolds, despite the anti-fibrotic effects of metformin.

Cell viability on the scaffolds was observed on days three and seven, by fluorescence microscopy of the scaffolds. Figure 9 shows the nuclei of DAPI-stained fibroblasts. Fluorescent images supported the fact that cell viability in all 3D-printed scaffolds was higher on day seven as compared to the day three. According to the images, PCL/PVA scaffolds had the highest number of viable cells on day three, whereas PCL/PVA-Met/PCL had the highest on day seven. During the cell culture, the fibroblasts on the scaffolds had the typical fibroblast morphology.

The cell viability below 30% is considered toxic according to ISO standard 10993-5. Cell viability results in this study were over 65% for all 3D-printed scaffolds. Therefore, it can be corroborated from the results that 3D-printed scaffolds were able to provide an environment for the cells to grow, thus establishing its applicability in tissue engineering applications. These scaffolds can also be used as a potential drug delivery system to enhance wound healing by not triggering excessive fibroblast proliferation.

## 4. Conclusions

Earlier studies demonstrated that sandwich structure could improve porosity and mechanical strength and avoid bursts of drug release, but these sandwich scaffolds were mainly fabricated using electrospinning. In this study, pure and metformin-loaded PCL/PVA and PCL/PVA/PCL sandwich scaffolds were successfully designed, and 3D printed, presenting a novelty to the literature. In addition to its antihyperglycemic effect, metformin was chosen to enhance diabetic wound healing due to its anti-inflammatory, anti-fibrotic, and antioxidant characteristics. Three-dimensional-printed scaffolds showed optimal open pore architecture and morphological properties to provide the necessary mechanical support and nutrient exchange to the neo-tissue, having average pore sizes varying from 63.6 ± 4 to 112.9 ± 3 μm. Owing to the sandwich structure, PCL/PVA/PCL scaffolds had a lower pore size than PCL/PVA scaffolds. Structural difference and drug loading contributed a significant difference in terms of degradation of the scaffolds. The addition of metformin caused an increase in degradation capability in both the PCL/PVA and PCL/PVA/PCL sandwich scaffolds. The highest degradation occurred in the PCL/PVA-Met scaffold because of its inherent design and influence of metformin. The sandwich structure could be an alternative to crosslinking scaffolds to prevent burst release. In vitro cell-based assessments established that the scaffolds were not cytotoxic to the cells. Based on the observations of this study, it might be concluded that metformin-loaded PCL/PVA and PCL/PVA/PCL scaffolds would increase the bioavailability and half-life of metformin and offer promising potential in diabetic wound healing.

## Figures and Tables

**Figure 1 polymers-14-05306-f001:**
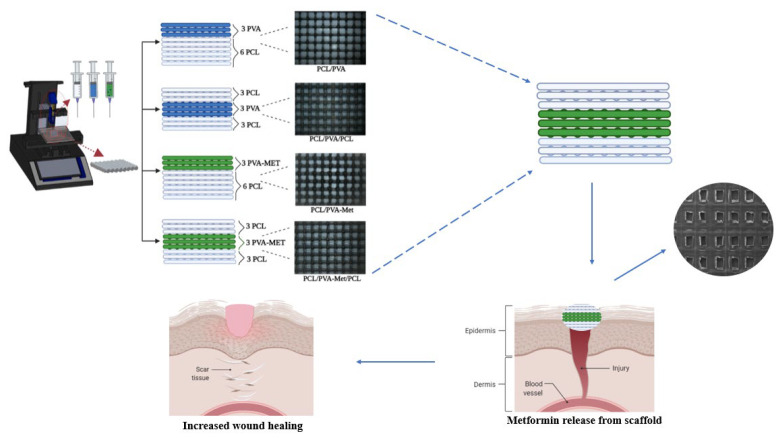
Production and characterization of the scaffolds using 3D printing and demonstration of their role in the wound healing process.

**Figure 2 polymers-14-05306-f002:**
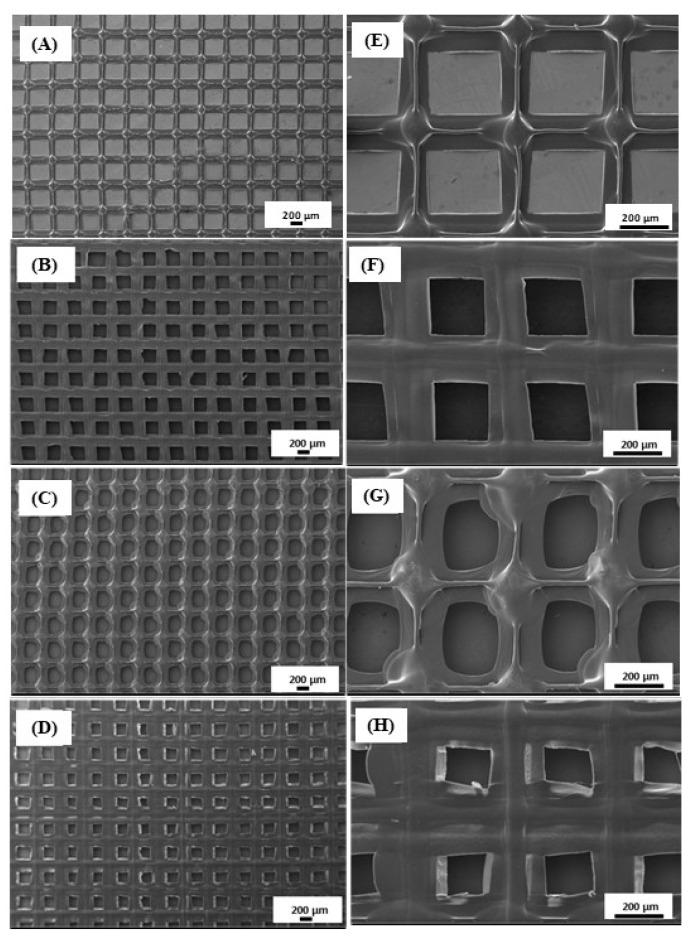
Scanning electron microscopy (SEM) images of the morphology of the scaffolds: PCL/PVA (**A**,**E**), PCL/PVA/PCL (**B**,**F**), PCL/PVA-Met (**C**,**G**), and PCL/PVA-Met/PCL (**D**,**H**).

**Figure 3 polymers-14-05306-f003:**
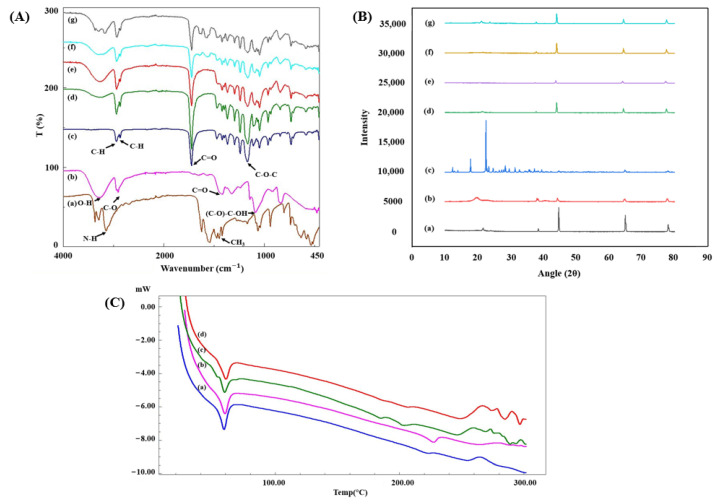
FTIR spectra of metformin ((**A**)), a), PVA ((**A**), b), PCL ((**A**), c), PCL/PVA ((**A**), d), PCL/PVA/PCL ((**A**), e), PCL/PVA-Met ((**A**), f), and PCL/PVA-Met/PCL ((**A**), g) scaffolds. XRD results of the PCL ((**B**), a), PVA ((**B**), b), metformin ((**B**), c) and PCL/PVA ((**B**), d), PCL/PVA/PCL ((**B**), e), PCL/PVA-Met ((**B**), f), and PCL/PVA-Met/PCL ((**B**), g) scaffolds; DSC curves of PCL/PVA ((**C**), a), PCL/PVA/PCL ((**C**), b), PCL/PVA-Met ((**C**), c), and PCL/PVA-Met/PCL ((**C**), d) scaffolds.

**Figure 4 polymers-14-05306-f004:**
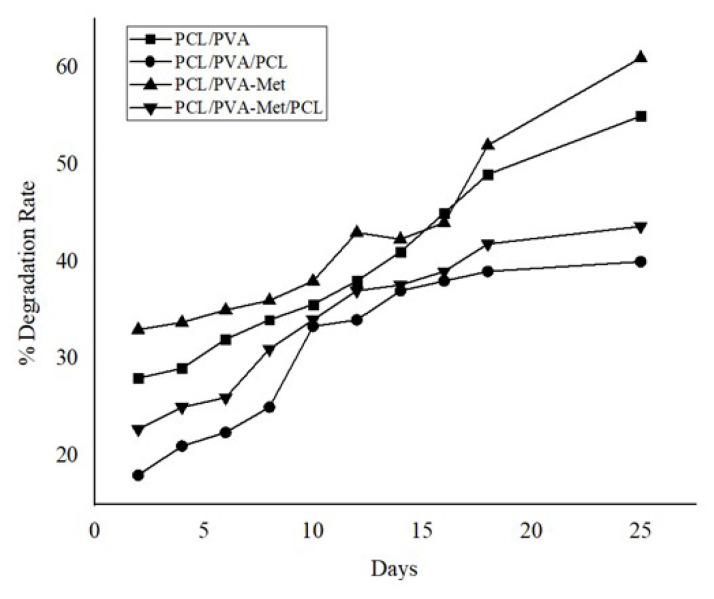
Degradation rates of 3D-printed scaffolds.

**Figure 5 polymers-14-05306-f005:**
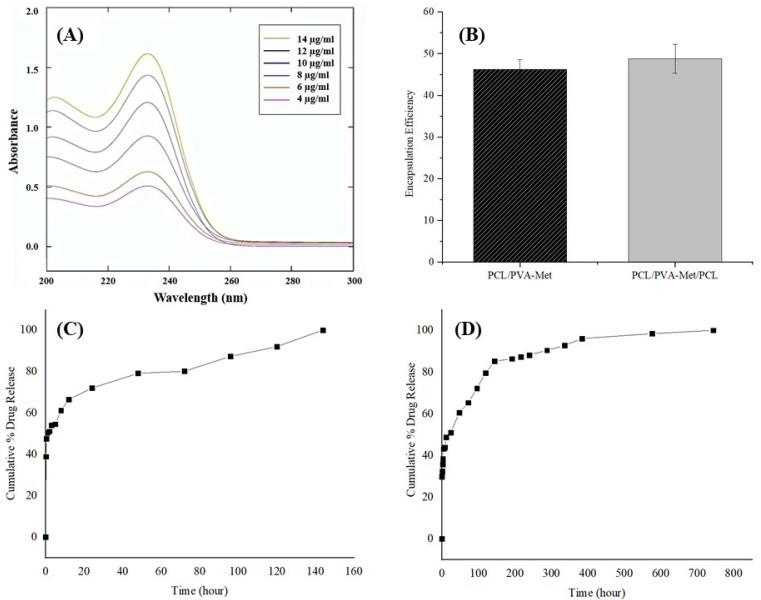
In vitro drug release of scaffolds: (**A**) Absorption spectra of metformin at 6 different concentrations, (**B**) encapsulation efficiency of the scaffolds, (**C**) release profiles of metformin from PCL/PVA-Met and (**D**) PCL/PVA-Met/PCL scaffolds.

**Figure 6 polymers-14-05306-f006:**
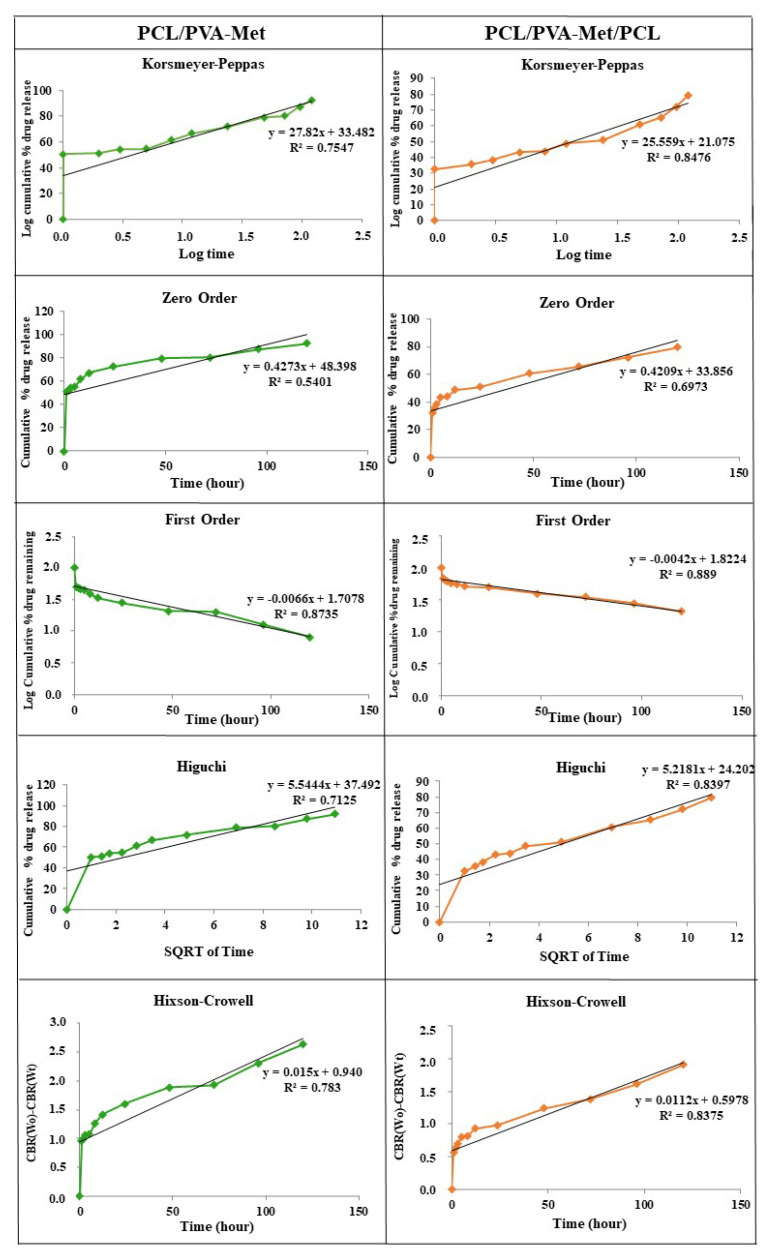
In vitro release kinetic models of the PCL/PVA-Met and PCL/PVA-Met/PCL scaffolds (please see Table 2 for further details).

**Figure 7 polymers-14-05306-f007:**
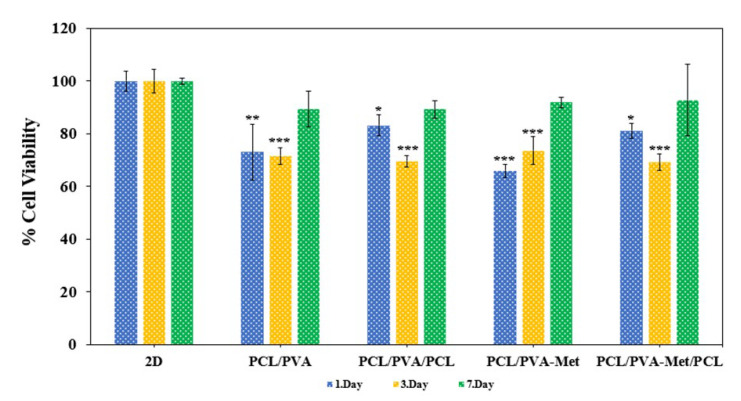
In vitro cell viability analysis using MTT assay of the fabricated scaffolds (* *p* < 0.05, ** *p* < 0.01, *** *p* < 0.001).

**Figure 8 polymers-14-05306-f008:**
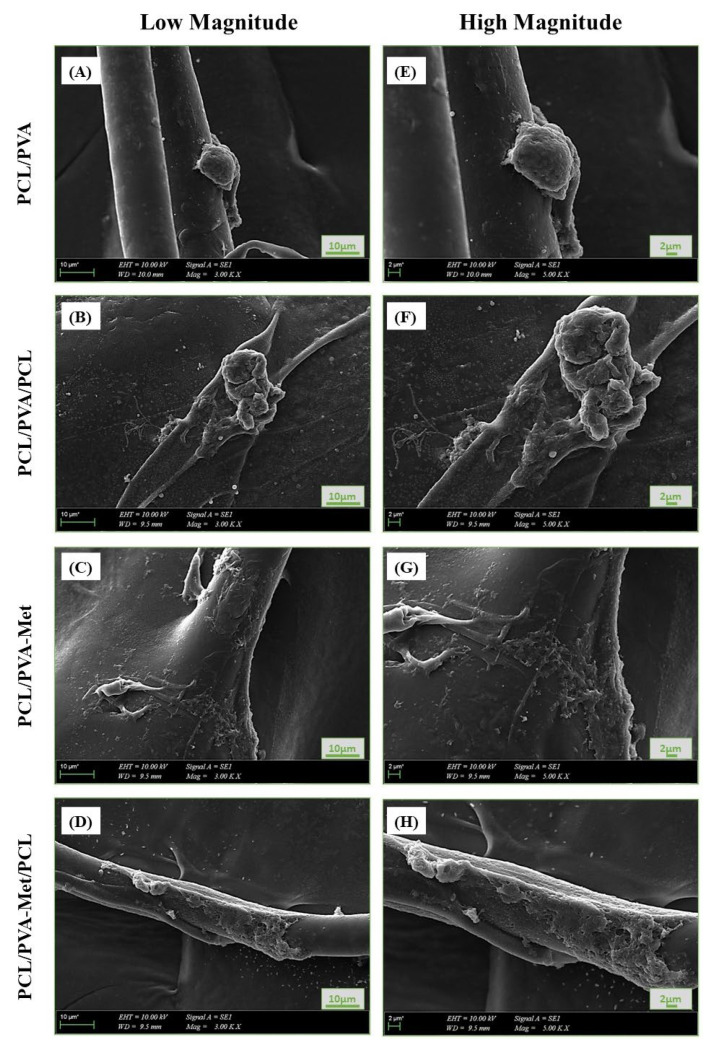
Morphology of the fibroblast cells attached on the scaffolds post 7 day incubation: SEM images of (**A**,**E**) PCL/PVA, (**B**,**F**) PCL/PVA/PCL, (**C**,**G**) PCL/PVA-Met, and (**D**,**H**) PCL/PVA-Met/PCL.

**Figure 9 polymers-14-05306-f009:**
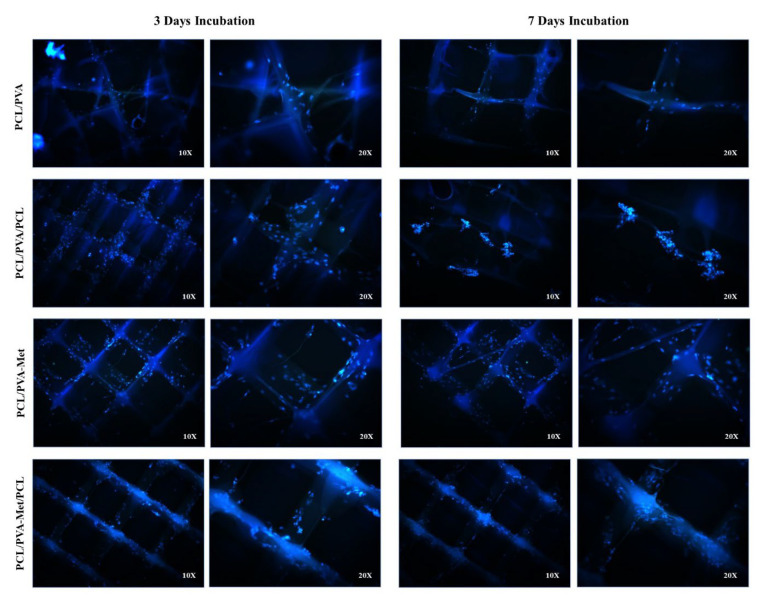
Fluorescent microscopy images of the DAPI stained fibroblast cells attached to the hybrid scaffolds, days 3 and 7 post-cell culture.

**Table 1 polymers-14-05306-t001:** Composition of the polymer solutions.

Polymer Solutions with Different Concentrations	PCL Content (Wt. %)	PVA Content (Wt. %)	Metformin Content (Wt. %)
PCL/PVA	20	20	0
PCL/PVA/PCL	20	20	0
PCL/PVA-Met	20	20	3
PCL/PVA-Met/PCL	20	20	3

**Table 2 polymers-14-05306-t002:** In vitro drug release kinetics of the PCL/PVA-Met and PCL/PVA-Met/PCL scaffolds.

	Korsmeyer–Peppas		Zero-Order		First-Order		Higuchi		Hixson–Crowell	
Sample	R^2^	n	R^2^	K_0_	R^2^	K_1_	R^2^	K_h_	R^2^	K_hc_
PCL/PVA-Met	0.7547	27.82	0.5401	0.4273	0.8735	−0.0066	0.7125	5.5444	0.7827	0.0149
PCL/PVA-Met/PCL	0.8476	25.559	0.6973	0.4209	0.889	−0.0042	0.8397	5.2181	0.8375	0.0112

**Table 3 polymers-14-05306-t003:** Diffusion mechanisms of the drug according to the diffusion constant (n) value.

Diffusion Constant (n)	Transport Mechanisms
0.45 ≤ n	Fickian diffusion mechanism
0.45 < n < 0.89	Non-Fickian transport
n = 0.89	Case II (relaxational) transport
n > 0.89	Super case II transport

## Data Availability

All data are contained within the Appendix A and the article.

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
