# Peer review of "Production of 3D Printed Bi-Layer and Tri-Layer Sandwich Scaffolds with Polycaprolactone and Poly (vinyl alcohol)-Metformin towards Diabetic Wound Healing"

_polymers, 2022, doi:10.3390/polym14235306_

Round 1

Reviewer 1 Report

Comments to authors are listed below:

  • The abstract lacks to present the numerical values from significant findings in this paper.

·         The characterisations and the applications Poly (vinyl alcohol, PVA) should be reported in detail and clearly, enhancing the information by including related recent literature. So, further recent references are listed below, should be considered.

https://link.springer.com/article/10.1007/s10924-019-01470-7

https://link.springer.com/article/10.1007/s42452-019-1111-2

https://journals.sagepub.com/doi/full/10.1177/0892705718804585

https://www.tandfonline.com/doi/abs/10.1080/15421406.2021.1946985

·         The novelty of this work should be reported clearly in the last paragraph of introduction section.

·         The tensile tests should reported in detail with including tensile stress-strain curve to verify the tensile strength, modulus, and strain at break characterisations instead of including tensile strength data in Figure 4.

Author Response

point by point response to all reviewers is attached here 

Reviewer 2 Report

This manuscript described an advanced method to fabricate four hybrid scaffolds used for diabetic wound healing. In the present work, PCL/PVA, PCL/PVA/PCL, and Metformin-loaded, PCL/PVA-Met and PCL/PVA-Met/PCL hybrid scaffolds with different designs were made and assessed their structure and related properties like molecular and chemical interaction, mechanical, thermal, swelling, and degradation analysis. The author team also evaluated the drug release rate and cytocompatibility of their scaffolds.

Overall, the research strategy is clear and the structure of the manuscript is straightforward. However, the quality of the data and the discussion about the obtained results need to be further improved and modified. Some concerns/questions below arise from the reading of the paper, especially on the data analysis and method introduction. Please address them in the revised work for the publication of this manuscript in Polymers

•           Details are not clearly disclosed in Section 2 Materials and Methods

•           In the presented work, the authors introduced bi-layered and three-layered structures. How well attached are those layered structures? There could be the possibility of delamination? The delamination potential has been tested/investigated? Please, provide evidence about this point.

•           Section 2.2.2, what’s the resolution of the extrusion nozzle? What’s the designed diameter of those patterns? Does your designed diameter of those patterns match the actual value?

•           Line 172, why does design 1/3 (Bi-layer type) have obviously different thickness ranges than design 2/4 (Tri-layer type)? The total layers for all 4 designs should be the same, based on the information disclosed in section 2.2.2. Please explain that.

•           Line 178: initial dry weights measurement. Are you measure the weight of three specimens for one type of scaffold, or measure one sample three times for the dry weight? Please clarify that

•           Line 179, just out of curiosity, do the scaffolds float or sink in the PBS?

•           Section 2.2.9, Is the Wd (initial dry weights) the same as the W0 (initial weights)? If same, please use the same term. If not, please explain the difference between those two terms.

•           Line 178: “initial dry weights (Wd) of each type of scaffolds were measured in triplicates”, do you mean one sample was measured three times, or do you have three specimens for one each type for this test? Please clarify that.

•           Equation (1), why does the author divides the Wt for the degradation index? In theory, the weight difference should be divided by W0 for the index.

•           Line 194, how long time does this shaking procedure last?

•           Line 197: What’s the frequency of this test?  Does it take every day and last for a month?

•           Line 198-204: How to guarantee that the drug is completely dissolved in the organic solvents, and if EE% is less than 95%, what’s the reason to lead the mass of the actual drug loaded is lighter than the designed load amount?

•           Section 2.2.11, which drug release kinetic model will you use in the presented study?

•           Line 253:  The authors reported the viscosity of three solutions under the shear rate 50 s-1. And do those values reflect the actual viscosity during the 3D printing? Have you done an investigation on the relationship between viscosity and the final dimension of the patterns? The reviewer cannot get the point of how those viscosity values impact the quality of your scaffolds. In my point, they’re a group of data, but not directly correlated to the quality or dimension of the scaffolds.

•           Paragraphs about surface morphology: It is recommended that you report your own data first, then discuss the impact or other scholars’ work. In a simple way, you’re recommended to swap those two paragraphs

•           Line 275-Line 280: Q1: Which cross-section of the scaffolds do you choose for characterizing the pore size? I assume that you characterize the surface morphology and based on your Figure 1, those layers are the PVA layer, PCL layer, PVA-MET layer and PCL layer, respectively. Then Q2: Why the values of Design 2 and Design 4 are quite different? They’re supposed to be the same type of PCL layers. Please explain that. And Q3, what’s the reason to explain the pore size decrease from 82.9 (design 3) to 63.6 (design 4)?  If you compare PVA-MET layer and PCL layer to get that conclusion, it doesn’t make sense. Please address these concerns, list the detailed layer you captured using SEM, and make a proper comparison.

•           Line 282: How do you define the term “homogeneity”? All your pore size values with a similar std around 2.7~4 μm. From the statistical view, all of them are homogeneous with a similar pore size range for each type of scaffold. Please address that. And also, the reason for “high crystalline structure” is not solid or proper. If you compare design 2 and design 4, both of them have 3 PCL layers on the top but their values are different (84.1 vs 63.6).

•           Figure 3: Where’re your 3A,f and 3A,g curves? It seems your statement from Line 306 to Line310 needs to be supported by those two curves. And also, the details of 3C are needed to disclose in the caption.

•           Line 334: Do you have the DSC curve of PCL? You may want to provide a comparison between pure PCL and the scaffolds you fabricated in this work, to show the impact of “inherent composite structure”. And also, it’s better to have pure PCL, PVA, and Metformin tested under the same conditions used for 4 scaffolds and displayed in the Figure 3C

•           Line 372-374: The data in Figure 5A does not support your claim: PCL/PVA-Met Scaffold (with a symbol of a triangle) DID NOT have the highest wet weight value, either on Day 1 or on Day 7.  And also, how do you characterize the water adsorption capacity? Please address this method in Section 2 and also correct the claim with the proper data.

•           Line 408: In section 2.2.10, you mentioned that all those measurements were made three times but from the data value I cannot see any standard deviation from the multi-measurements.

•           Figure 6C and D: It is strongly recommended the authors combine Figure 6C and D so it’s easier to compare the different release profiles from two scaffolds

•           Line 415: You need to swap Figure 6C and D? From your statement in line 415, Figure C should be the right plot for PCL/ PVA-met/PCL scaffold.

•           Line 470: Do you have the cell growth ratio for 4 scaffolds and the control sample on different days?

Author Response

(The authors gave the same response as above.)

Reviewer 3 Report

This manuscript demonstrated a biocompatible sandwich scaffolds with multi-layers polycaprolactone and poly (vinyl alcohol)-metformin by 3D printing technology. This strategy offers an alternative to enhance diabetic wound healing as a promising drug delivery systems. The architecture and representation of this paper are relatively complete and the meaningful. It is recommended to accept publication after minor revision.

1. The authors just use DAPI staining. My advice is that the biocompatibility of the sandwich scaffolds should be performed with characterization with calcein-AM/PI kit for live-dead staining or Calcein-AM/hochesst 33342 for living staining. 

Author Response

(The authors gave the same response as above.)

Round 2

Reviewer 1 Report

No comments. 

Author Response

(The authors gave the same response as above.)

Reviewer 2 Report

Thanks for the authors’ response. The reviewer is okay with most of the answers and responses. But two major problems are not answered properly: 1. The pore size and the FILAMENT dimension: the reviewer expects to see a convincing answer to explain the pore size change and how Met impact the SURFACE pore size. 2. The wet weight calculation: The interpretation and explanation in the original and revised manuscript are not convincing. Some statements even contradict the plot data (Fig 4A). Please be cautious and make a proper response.

Please address them in your next version of the manuscript, highlighting the changes/explanations.

1.     Section 2.2.2, what’s the resolution of the extrusion nozzle? What’s the designed diameter of those patterns? Does your designed diameter of those patterns match the actual value?

Response: The diameter of the nozzle tip was 0.15 mm.

We printed square geometry, with a dimension 20 mm x 20 mm x 1 mm. The designed patterns match the printed dimensions with allowable tolerances for 3D printed polymer structures.

The filament diameter of the PCL/PVA scaffold was 0.065±3.66 mm, PCL/PVA/PCL scaffold was 0.110±6.25 mm, PCL/PVA-Met scaffold was 0.107±4.43, and PCL/PVA-Met/PCL scaffold was 0.112±6.98 mm. So, mostly the actual diameters of 3D printed scaffolds were matched with the design.

Reply to the authors’ response: Your filament diameter (0.065-0.112mm, with huge std?? 3.66/6.25/4.43/6.98) is smaller than the nozzle’s resolution of 0.15mm, is it possible to provide a matched fabrication? Does the diameter of the nozzle limit the filament diameter not be lower than 0.15mm? How do you control the extrusion amount?

2.     Line 275-Line 280: Q1: Which cross-section of the scaffolds do you choose for characterizing the pore size? I assume that you characterize the surface morphology and based on your Figure 1, those layers are the PVA layer, PCL layer, PVA-MET layer and PCL layer, respectively. Then Q2: Why the values of Design 2 and Design 4 are quite different? They’re supposed to be the same type of PCL layers. Please explain that. And Q3, what’s the reason to explain the pore size decrease from 82.9 (design 3) to 63.6 (design 4)?  If you compare PVA-MET layer and PCL layer to get that conclusion, it doesn’t make sense. Please address these concerns, list the detailed layer you captured using SEM, and make a proper comparison.

Response Q1: SEM images used to analyse average pore sizes were the top view of the scaffolds.

Reply to the authors’ response: If your information is correct, then you will see PCL layers only in Design 2 and Design 4 since the surfaces of both designs are wrapped by PCL layers. Then that’s my Q2, they’re the same type of PCL layers, and why did the pore size change from 84.1 to 63.4? Considering their small std, that’s a significant difference.

Response Q2: The 4 different images in Figure 2 represent 4 different scaffolds. In Figure 2 (A, E) there is a PCL layer at the bottom and a PVA layer on top, in Figure 2 (B, F) there is a PCL layer above and below the middle PVA layer. In Figure 2 (C, G) there is a PCL layer at the bottom and a PVA-Met layer on top, while in Figure 2 (D, H) there is a PCL layer above and below the middle PVA-Met layer.

Reply to the authors’ response: the authors didn’t answer this question. If you measured PCL layers for Design 2 (B,F) and Design 4 (D,H), why their values are different? Again, based on your clarification in Q1, they should be the same PCL layers

Response Q3: Designs 2 and 4 which are represented in Figure 2 (B, F) and Figure 2 (D, H), respectively are the same. The only difference is the presence of Metformin in the PVA solution. For PCL/PVA/PCL scaffolds the average pore size was 84.1±2.7 μm. Metformin addition decreased the average pore size to 63.6±4.0 μm for PCL/PVA-Met/PCL scaffolds. As stated in the study, the addition of drugs to the scaffolds may cause a decrease in pore size. Similarly, in this study, a decrease in the pore size of the scaffolds was observed when Metformin was added.

Reply to the authors’ response: Based on your printing tech, Metformin addition should only impact the PVA layers and you do not add Metformin in PCL layers. Unless the metformin can impact the PCL surface layer (and how?), the authors may not answer properly about the reason why Design 4’s pore size (the PCL layer’s pore size) is smaller than Design 2’s pore size (the PCL layer’s pore size)

And the authors also claim that “but the sandwich scaffolds' homogeneity was higher as compared to the others;” If you just compare PVA top layers (Design 1), PCL top layers (Design 2), PVA-Met top layers (Design 3), PCL top layers (Design 4), that’s no wonder that Design 2 and 4 look better since you just check the PCL layers.

3.     Line 282: How do you define the term “homogeneity”? All your pore size values with a similar std around 2.7~4 μm. From the statistical view, all of them are homogeneous with a similar pore size range for each type of scaffold. Please address that. And also, the reason for “high crystalline structure” is not solid or proper. If you compare design 2 and design 4, both of them have 3 PCL layers on the top but their values are different (84.1 vs 63.6).

Response: The term homogeneity refers that there is no significant difference in the average pore sizes of the different scaffolds. The standard deviation between pore sizes was small.

The high crystalline structure of PCL used here due to the presence of PCL in the upper layer, which is absent in PCL/PVA and PCL/PVA-Met scaffolds in PCL/PVA/PCL and PCL/PVA-Met/PCL scaffolds.

In Designs 2 and 4, the presence of PCL in the lower and upper layers is the same, but Metformin added to the PVA solution may have changed the pore size. As mentioned in lines 267-268 with the reference number [42], the addition of drugs to the scaffolds reduced the pore sizes. In our study, the pore sizes may have decreased due to the interaction between the drug and the polymer solution. This interaction increased the viscosity of the drug-loaded polymer solution therefore the pore size of the scaffold was decreased.

Reply to the authors’ response: I understand your point that Metformin added to the PVA solution may have changed the pore size (of PVA layers). But my point is that will the Metformin added impact the PCL layers’ porosity? And if it is, how? You didn’t add Metformin to PCL, correct? All the pore size calculation seems based on measurements of SEM images, which means you calculate the pore size of PCL layers (for Design 2 and Design 4). Without solid evidence/reference to clarify how Met impact the SURFACE layers of each SEM image, it’s hard to be convinced based on your explanation.

4.     Line 372-374: The data in Figure 5A does not support your claim: PCL/PVA-Met Scaffold (with a symbol of a triangle) DID NOT have the highest wet weight value, either on Day 1 or on Day 7.  And also, how do you characterize the water adsorption capacity? Please address this method in Section 2 and also correct the claim with the proper data.

Response: The authors thankfully acknowledge the comments of the reviewer. Necessary corrections have been made in the lines between 354-367 as mentioned below.

“Accordingly, while PCL/PVA-Met/PCL scaffold had the highest wet weight value on day 1 and PCL/PVA scaffold had the lowest value. PCL/PVA-Met/PCL sandwich scaffold had higher initial water absorption capacity. It can be attributed that high pore sizes could lead to increased water uptake capacity.         

Reply to the authors’ response: “. PCL/PVA-Met/PCL sandwich scaffold had higher initial water absorption capacity. It can be attributed that high pore sizes could lead to increased water uptake capacity.” Are you sure about this explanation? The pore size of Design 4 is the smallest in your previous result and now in this section, it has the higher pore sizes.

      In contrast, the upper hydrophobic PCL layer in the PCL/PVA/PCL and PCL/PVA-Met/PCL sandwich scaffolds might have resulted in decreased water absorption capacity on day 7.”

Reply to the authors’ response: The response above is not convincing. Here’re two major questions: 1) During those 7 days, your PVA and PVA-Met will degrade quickly and your PCL will degrade slowly (based on their biodegradable nature and rate), how do you eliminate those weight loss when you calculate water absorption capacity? 2) How do you calculate the amount of water absorption on Day 7? From Fig 4a, on Day 7, two sandwich scaffolds exhibited quite different weight wet (the highest and the lowest) and they cannot be explained as “resulted in decreased water absorption capacity”.

The characterisation of the water uptake capacity was mentioned in the 2.2.8 part of the material method section of the revised manuscript.

I didn’t see your introduction about how to conduct a water uptake capacity test. Please use the red font to highlight in Section 2.2.8

To summarize, I don’t quite believe the significance of the data in this section, unless the authors could eliminate the factor from the weight loss. And also, the author’s explanation contradicts the figure data. (Day 1’s high pore size explanation and Day 7’s PCL’s impact) 

Author Response

(The authors gave the same response as above.)

Round 3

Reviewer 2 Report

The author team has tried their best to address the comments and the reviewer's concerns in this round. For some topics which cannot be addressed in this draft (and the author team removed those sections in the revised work), please design your experiment properly in the future and keep investigating them.